# Core motifs predict dynamic attractors in combinatorial threshold-linear networks

**Caitlyn Parmelee** [1], **Samantha Moore** [2], **Katherine Morrison**[3]�, **Carina Curto** [4]� *

1 Mathematics Department, Keene State College, Keene, NH, United States of America, 2 Department of Mathematics, University of North Carolina at Chapel Hill, State College, PA, United States of America, 3 School of Mathematical Sciences, University of Northern Colorado, Greeley, CO, United States of America, 4 Department of Mathematics, The Pennsylvania State University, University Park, PA, United States of America

055 These authors contributed equally to this work.
* ccurto@psu.edu

**Data Availability Statement:** All relevant data are within the paper and its Supporting information files.

**Funding:** This work was supported by NIH R01 EB022862 (CC & KM), NIH R01 NS120581 (CC),

## Abstract

Combinatorial threshold-linear networks (CTLNs) are a special class of inhibition-dominated TLNs defined from directed graphs. Like more general TLNs, they display a wide variety of nonlinear dynamics including multistability, limit cycles, quasiperiodic attractors, and chaos. In prior work, we have developed a detailed mathematical theory relating stable and unstable fixed points of CTLNs to graph-theoretic properties of the underlying network. Here we find that a special type of fixed points, corresponding to *core motifs*, are predictive of both static and dynamic attractors. Moreover, the attractors can be found by choosing initial conditions that are small perturbations of these fixed points. This motivates us to hypothesize that dynamic attractors of a network correspond to unstable fixed points supported on core motifs. We tested this hypothesis on a large family of directed graphs of size n = 5, and found remarkable agreement. Furthermore, we discovered that core motifs with similar embeddings give rise to nearly identical attractors. This allowed us to classify attractors based on structurally-defined graph families. Our results suggest that graphical properties of the connectivity can be used to predict a network's complex repertoire of nonlinear dynamics.

## Introduction

The vast majority of the literature on attractor neural networks has focused on fixed point attractors. The typical scenario is that of a network that contains either a discrete set of stable fixed points, as in the Hopfield model, or a continuum of marginally stable fixed points, as in continuous attractor networks. Depending on initial conditions, or in response to external inputs, the activity of the network converges to one of these fixed points. These are sometimes referred to as *static* attractors, because the fixed point is in an equilibrium or steady state. But neural networks, even very simple ones like threshold-linear networks (TLNs), can also exhibit *dynamic* attractors with periodic, quasiperiodic, or even chaotic orbits. What network

NSF DMS-1951165 (CC), and NSF DMS-1951599 (KM). The funders had no role in study design, data collection and analysis, decision to publish, or preparation of the manuscript.

**Competing interests:** The authors have declared that no competing interests exist.

architectures support these more complex attractors? And can their existence be predicted based on the structure of the underlying connectivity graph?

In this work, we tackle these questions in the context of combinatorial threshold-linear networks (CTLNs), which are a special class of TLNs defined from directed graphs. TLNs have been widely used in computational neuroscience as a framework for modeling recurrent neural networks, including associative memory networks [1–8]. And CTLNs are a subclass that are especially tractable mathematically [9–12]. What graph structures support dynamic attractors in CTLNs?

We begin by observing an apparent correspondence between a network's *minimal* fixed points and its attractors. This leads us to hypothesize that fixed points supported on a special class of subgraphs, called *core motifs*, are predictive of both static and dynamic attractors. Next, we test this hypothesis on a large family of CTLNs on small graphs of $n = 5$ nodes. We find that, with few exceptions, fixed points supported on core motifs correspond precisely to the attractors of the network. Moreover, we find that core motifs with similar embeddings in the larger network give rise to nearly identical attractors. This enables us to classify the observed attractors and identify common structural properties of the graphs that support them. Our results illustrate how the structure of a network can be used to predict the emergence of various attractors by way of graphical analysis.

## Methods and models

We study dynamic attractors in a family of threshold-linear networks (TLNs). The firing rates $x_1(t), \ldots, x_n(t)$ of $n$ recurrently-connected neurons evolve in time according to the standard TLN equations:

$$\frac{dx_i}{dt} = -x_i + \left[ \sum_{j=1}^{n} W_{ij} x_j + b_i \right]_+, \quad i = 1, \ldots, n \tag{1}$$

where $n$ is the number of neurons. The dynamic variable $x_i(t) \in \mathbb{R}_{\geq 0}$ is the activity level (or "firing rate") of the $i^{\text{th}}$ neuron, and $b_i$ can represent a threshold or an external input. The values $W_{ij}$ are entries of an $n{\times}n$ matrix of real-valued connection strengths. The threshold nonlinearity $[\cdot]_+ \overset{\text{def}}{=} \max\{0, \cdot\}$ is critical for the model to produce nonlinear dynamics; without it, the system would be linear.

### CTLNs

Combinatorial threshold-linear networks (CTLNs) are a special case of inhibition-dominated TLNs, where we restrict to having only two values for the connection strengths $W_{ij}$. These are obtained as follows from a directed graph $G$, where $j \to i$ indicates that there is an edge from $j$ to $i$ and $j \not\to i$ indicates that there is no such edge:

$$W_{ij} = \begin{cases} 0 & \text{if } i = j, \\ -1 + \varepsilon & \text{if } j \to i \text{ in } G, \\ -1 - \delta & \text{if } j \to i \text{ in } G. \end{cases} \tag{2}$$

Additionally, CTLNs typically have a constant external input $b_i = \theta$ in order to ensure the dynamics are internally generated rather than inherited from a changing or spatially heterogeneous input. A CTLN is thus completely specified by the choice of a graph $G$, together with three real parameters: $\varepsilon$, $\delta$, and $\theta$. We additionally require that $\delta > 0$, $\theta > 0$, and

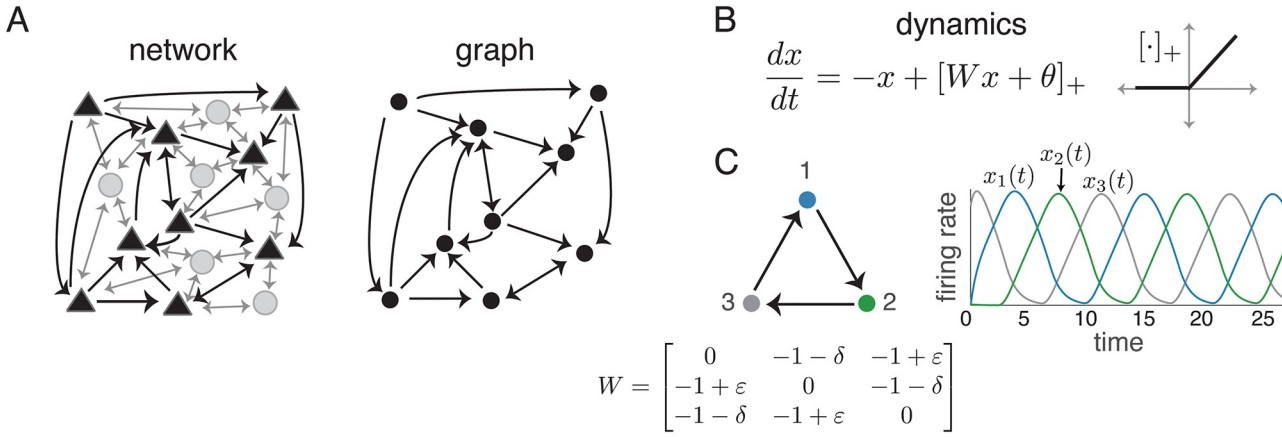

**Fig 1. CTLNs.** (A) A neural network with excitatory pyramidal neurons (triangles) and a background network of inhibitory interneurons (gray circles) that produces a global inhibition. The corresponding graph (right) retains only the excitatory neurons and their connections. (B) TLN dynamics and the graph of the threshold-nonlinearity $[\cdot]_+ = \max\{0, \cdot\}$. (C) A graph that is a 3-cycle (left) and its corresponding CTLN matrix $W$. (Right) A solution for the corresponding CTLN, with parameters $\varepsilon = 0.25$, $\delta = 0.5$, and $\theta = 1$, showing that network activity follows the arrows in the graph. Peak activity occurs sequentially in the cyclic order 123.

$0 < \varepsilon < \dfrac{\delta}{\delta + 1}$. When these conditions are met, we say the parameters are within the *legal range*. (The upper bound on $\varepsilon$ ensures that subgraphs consisting of a single directed edge $i \to j$ are not allowed to support stable fixed points [9].) Note that the upper bound on $\varepsilon$ implies $\varepsilon < 1$, and so the $W$ matrix is always effectively inhibitory. For fixed parameters, only the graph $G$ varies between networks. Unless otherwise noted, the simulations presented here have parameters $\theta = 1$, $\varepsilon = 0.25$, and $\delta = 0.5$. We will refer to these as the *standard parameters*.

We interpret the CTLN as modeling a network of $n$ excitatory neurons, whose net interactions are effectively inhibitory due to a strong global inhibition (Fig 1A). When $j \not\to i$, we say *j strongly inhibits i*; when $j \to i$, we say *j weakly inhibits i*. Note that because $-1 - \delta < -1 < -1 + \varepsilon$, when $j \not\to i$ neuron $j$ inhibits $i$ *more* than it inhibits itself via its leak term; when $j \to i$, neuron $j$ inhibits $i$ *less* than it inhibits itself. These differences in inhibition strength cause the activity to follow the arrows of the graph (see Fig 1C).

## Fixed points

Stable fixed points of a network are of obvious interest because they correspond to static attractors [3, 6, 7, 12]. One of the most striking features of CTLNs, however, is the strong connection between unstable fixed points and dynamic attractors [10]. This is our main focus here.

A fixed point $x^* \in \mathbb{R}^n$ of a TLN is a solution that satisfies $dx_i/dt|_{x=x^*} = 0$ for each $i \in [n]$, where $[n] \overset{\text{def}}{=} \{1, \ldots, n\}$. The *support* of a fixed point is the subset of active neurons, $\text{supp}\,x^* = \{i \mid x_i^* > 0\}$. CTLNs (and TLNs) are piecewise-linear dynamical systems, and we typically require a nondegeneracy condition that is generically satisfied and implies nondegeneracy in each linear regime [11]. As a result, we have that for a given network there can be at most one fixed point per support. Thus, we can label all the fixed points of a network by their support, $\sigma = \text{supp}\,x^* \subseteq [n]$. We denote this collection of supports by:

$$\text{FP}(G) = \text{FP}(G, \varepsilon, \delta) \overset{\text{def}}{=} \{\sigma \subseteq [n] \mid \sigma \text{ is a fixed point support of the associated CTLN}\}.$$

Note that once we know $\sigma \in \text{FP}(G)$ is the support of a fixed point, the fixed point itself is easily recovered. Outside the support, we must have $x_i^* = 0$ for all $i \notin \sigma$. Within the support, $x^*$ is

given by:

$$x_\sigma^* = \theta(I - W_\sigma)^{-1}1_\sigma,$$

where $W_\sigma$ is the induced submatrix obtained by restricting rows and columns to $\sigma$, and $1_\sigma$ is a column vector of all 1s of length $|\sigma|$.

A useful fact is that a fixed point for a CTLN with graph $G$ is also a fixed point for any sub-network containing its support. A *subnetwork* supported on $\sigma$ is a CTLN for the *induced sub-graph* $G|_\sigma$ obtained from $G$ by restricting to the vertices of $\sigma$ and keeping only edges $i \to j$ for $i$, $j \in \sigma$ (see [11]). In particular, if $\sigma \in$ FP($G$), then $\sigma \in$ FP($G|_\sigma$). The converse is not true: one can have $\sigma \in$ FP($G|_\sigma$) in the subnetwork, but the fixed point may *not survive* the embedding into the larger network, and so $\sigma \notin$ FP($G$).

## Graph rules

In prior work, a series of *graph rules* were proven that can be used to determine fixed points of a CTLN by analyzing the structure of the graph $G$ [11, 12]. These rules are all independent of the choice of parameters $\varepsilon$, $\delta$, and $\theta$. Some of the simplest graph rules are also quite powerful, and can be used to fully determine FP($G$) for many graphs. Note that these are only valid for *nondegenerate* CTLNs, a condition defined in [11] that generically holds.

Table 1 lists a few of these rules, which were all proven in [11]. To state the rules, we need some graph-theoretic terminology. A vertex $i$ of a graph $G$ is a *source* if it has no incoming edges $j \to i$, and it is a *proper source* if it also has at least one outgoing edge $i \to j$. A *sink* is a vertex with no outgoing edges. A graph is *uniform in-degree* with degree $d$ if all vertices receive exactly $d$ incoming edges (see Fig 2). Note that a vertex can be a source or a sink in an induced subgraph, $G|_\sigma$, but not in the full graph $G$. Similarly, a subgraph $G|_\sigma$ may be uniform in-degree even if the vertices of $\sigma$ do not have the same in-degree within the full graph $G$.

The sources rule implies that there can be no fixed points supported on a pair of vertices $\{i, j\}$ corresponding to a single directed edge, $i \to j$, since $i$ is a proper source in the induced sub-graph. The sinks rule tells us that the only singletons that can support fixed points correspond to sinks of $G$. The uniform in-degree rule implies that cycles, which have uniform in-degree $d = 1$, support fixed points if and only if there is no vertex outside the cycle receiving two or more edges from it. It also implies that cliques of size $m$, which have uniform in-degree $m - 1$, support fixed points if and only if they are *target-free*: that is, if and only if there is no vertex $k$ outside the clique receiving edges from all $m$ vertices of the clique. In fact, we have shown in [9] that target-free cliques, including sinks, correspond to *stable* fixed points. A third consequence of the uniform in-degree rule is that if $G|_\sigma$ is an independent set (uniform in-degree 0), then $\sigma \in$ FP($G$) if and only if each $i \in \sigma$ is a sink in $G$.

To see how these graph rules, together with parity, can be used to work out FP($G$), consider the graph in Fig 3A. From the sinks rule we immediately see that $\{4\} \in$ FP($G$), but no other singleton supports a fixed point. From the sources rules we see that none of the pairs $\{1, 2\}$, $\{2, 3\}$,

**Table 1. Graph rules for CTLNs.**

| rule name | graph rule (theorem) |
|---|---|
| sources | if $i \in \sigma$ is a proper source in $G|_\sigma$ or in $G$, then $\sigma \notin$ FP($G$) |
| sinks | if $\sigma = \{i\}$ is a singleton, then $\sigma \in$ FP($G$) iff $i$ is a sink of $G$ |
| uniform | if $G|_\sigma$ is uniform in-degree $d$, then |
| in-degree | $\sigma \in$ FP($G$) $\Leftrightarrow \forall k \notin \sigma$, $k$ receives at most $d$ edges from $\sigma$ |
| parity | the total number of fixed points, $|$FP($G$)$|$, is odd |

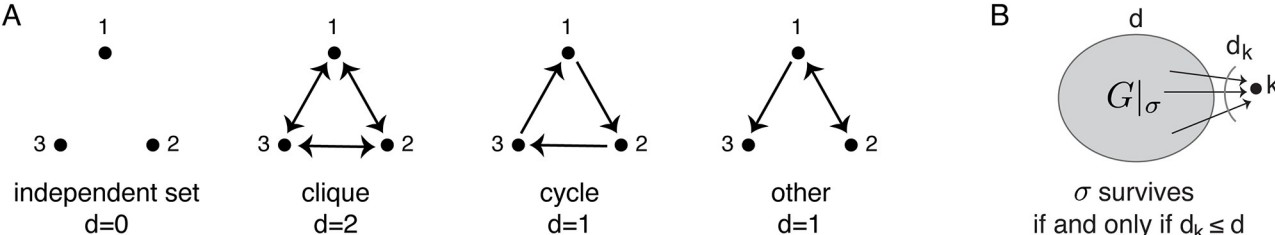

**Fig 2. Uniform in-degree graphs.** (A) All $n = 3$ graphs with uniform in-degree. (B) Cartoon showing survival rule for an arbitrary subgraph with uniform in-degree $d$.

{1, 3}, and {3, 4} are in FP($G$), as they correspond to edges having a proper source. Using the uniform in-degree rule we can also rule out the independent sets {1, 4} and {2, 4}, since 1 and 2 are not sinks. On the other hand, the uniform in-degree rule does imply that the 3-cycle {1, 2, 3} ∈ FP($G$). All other subsets of size three can be ruled out because they have a proper source in the induced subgraph $G|_\sigma$.

After checking all subsets of vertices of size 1, 2, or 3, we have found that only 4, 123 ∈ FP($G$). Here we are simplifying notation for fixed point supports by writing 4 instead of {4} and 123 instead of {1, 2, 3}. Now we can use parity to conclude that we must also have 1234 ∈ FP($G$), since this is the only remaining support and the total number of fixed points must be odd. Note that the rules we have used apply irrespective of the choice of $\varepsilon$, $\delta$ or $\theta$, provided they are in the legal range. It follows that FP($G$) = {4, 123, 1234} is invariant under changes of these parameters.

## Results

### Core fixed points and core motifs

The starting point for this work was the following remarkable observation about CTLNs: namely, that their fixed points appear to give rise to both *static* and *dynamic* attractors.

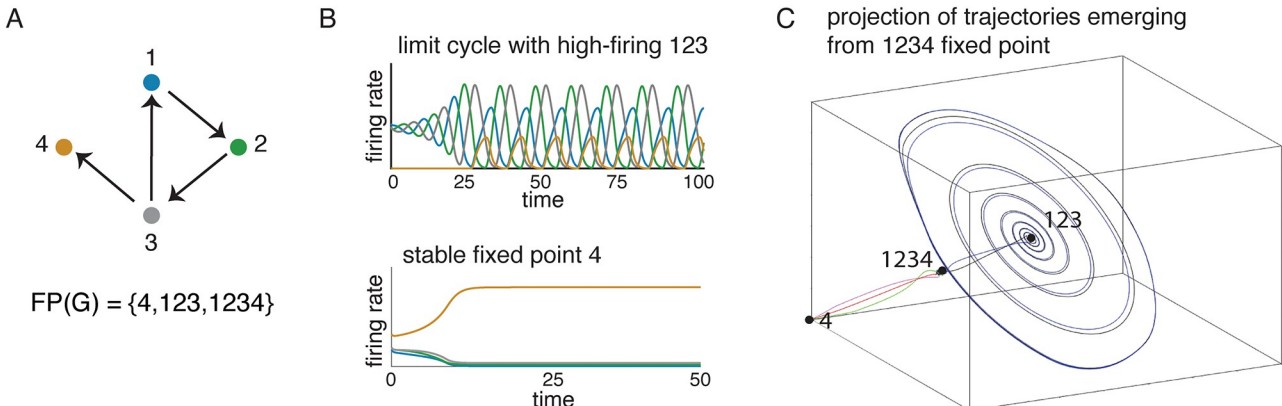

**Fig 3. An example CTLN and its attractors.** (A) The graph of a CTLN. Using graph rules, we can compute FP($G$). (B) Solutions to the CTLN with the graph in panel A using the standard parameters $\theta = 1$, $\varepsilon = 0.25$, and $\delta = 0.5$. (Top) The initial condition was chosen as a small perturbation of the fixed point supported on 123. The activity quickly converges to a limit cycle where the high-firing neurons are the ones in the fixed point support. (Bottom) A different initial condition yields a solution that converges to the static attractor corresponding to the stable fixed point on node 4. (C) The three fixed points are depicted in a three-dimensional projection of the four-dimensional state space. Perturbations of the fixed point supported on 1234 produce solutions that either converge to the limit cycle shown in panel B, or to the stable fixed point. This fixed point thus lives on the boundary of the two basins of attraction, and behaves as a "tipping point" between the two attractors.

Consider again the network in Fig 3A. From graph rules, we already saw that FP($G$) = {4, 123, 1234}, irrespective of the choice of $\varepsilon$, $\delta$ or $\theta$. The support 4 corresponds to a sink in the graph, and gives rise to a stable fixed point (i.e., a static attractor). Initial conditions that are small perturbations of this fixed point will result in the network activity converging back to it (Fig 3B, bottom). In contrast, the fixed point supported on 123 gives rise to a limit cycle (i.e., a dynamic attractor). Initial conditions near this fixed point result in activity that converges to a periodic trajectory whose high-firing neurons are 1, 2, and 3 (Fig 3B, top). The last fixed point, with full support 1234, does not have a corresponding attractor. Initial conditions that are small perturbations of this fixed point can either converge to the attractor for 4 or for 123 (Fig 3C). This fixed point lies on the boundaries of the two basins of attraction, and can thus be considered a "tipping point."

Which fixed points correspond to attractors, and which ones are tipping points? Fig 3 provides a hint: the fixed points giving rise to attractors have supports 4 and 123, which are *minimal* under inclusion in FP($G$). The last fixed point, 1234, has a support that contains smaller fixed point supports. We will call a fixed point *minimal* in $G$ if its support $\sigma$ is minimal in FP($G$).

Next, consider Fig 4A. Using the graph rules in Table 1, together with two additional rules in [11] (Lemma 21 and graphical domination), we can work out FP($G$) for this network as well. We find that there are two minimal fixed points, supported on 125 and 235. Consistent with our previous observations, each of these fixed points has a corresponding attractor. Specifically, we say that a fixed point *corresponds* to an attractor if

(i) initial conditions that are small perturbations from the fixed point lead to solutions that converge to the attractor, and

(ii) the high-firing neurons in the attractor match the support of the fixed point.

Just as we saw in Fig 3, the induced subgraphs $G|_\sigma$ corresponding to the minimal fixed point supports $\sigma$ = 125 and $\sigma$ = 235 are 3-cycles. Note, however, that the graph in Fig 4A actually has a third 3-cycle, 145, yet this one does not support a fixed point of $G$ and also has no corresponding attractor. The reason 145 $\notin$ FP($G$) is because node 3 receives two edges from 145, and so the uniform in-degree rule tells us the fixed point does not survive in the larger graph. In contrast, both 125 and 235 do have fixed points that survive to the full network. We see from this example that it is not enough to have a subgraph that supports an attractor. The 3-cycle whose fixed point does not survive the embedding has no corresponding attractor.

Does every minimal fixed point of $G$ have a corresponding attractor? Unfortunately, minimality is not enough. Fig 4B depicts a network built from the one in panel A by adding a single node, 6. Because 6 receives edges from both 2 and 5, it ensures that the 125 and 235 fixed points do not survive to the full network. As a result, 1235 also dies but the fixed point 1245 remains and becomes minimal. This fixed point does *not* have a corresponding attractor, however. Small perturbations of the fixed point for 1245 lead the network to converge to the attractor corresponding to the other minimal fixed point, 236. The main difference between these minimal fixed points is that 1245 is not minimal in its own subnetwork, $G|_{1245}$, while 236 is still minimal in $G|_{236}$. This motivates the following definition:

**Definition 1** (core fixed point). We say that a fixed point of a CTLN on a graph $G$ is a *core fixed point* if its support $\sigma \in$ FP($G$) is minimal (under inclusion) in FP($G$) and is also minimal in FP($G|_\sigma$).

Equivalently, $\sigma$ is the support of a core fixed point if and only if $\sigma \in$ FP($G$) and FP($G|_\sigma$) = {$\sigma$}. This is because the minimality of $\sigma$ in $G|_\sigma$ guarantees that $\sigma$ is the unique fixed point of $G|_\sigma$

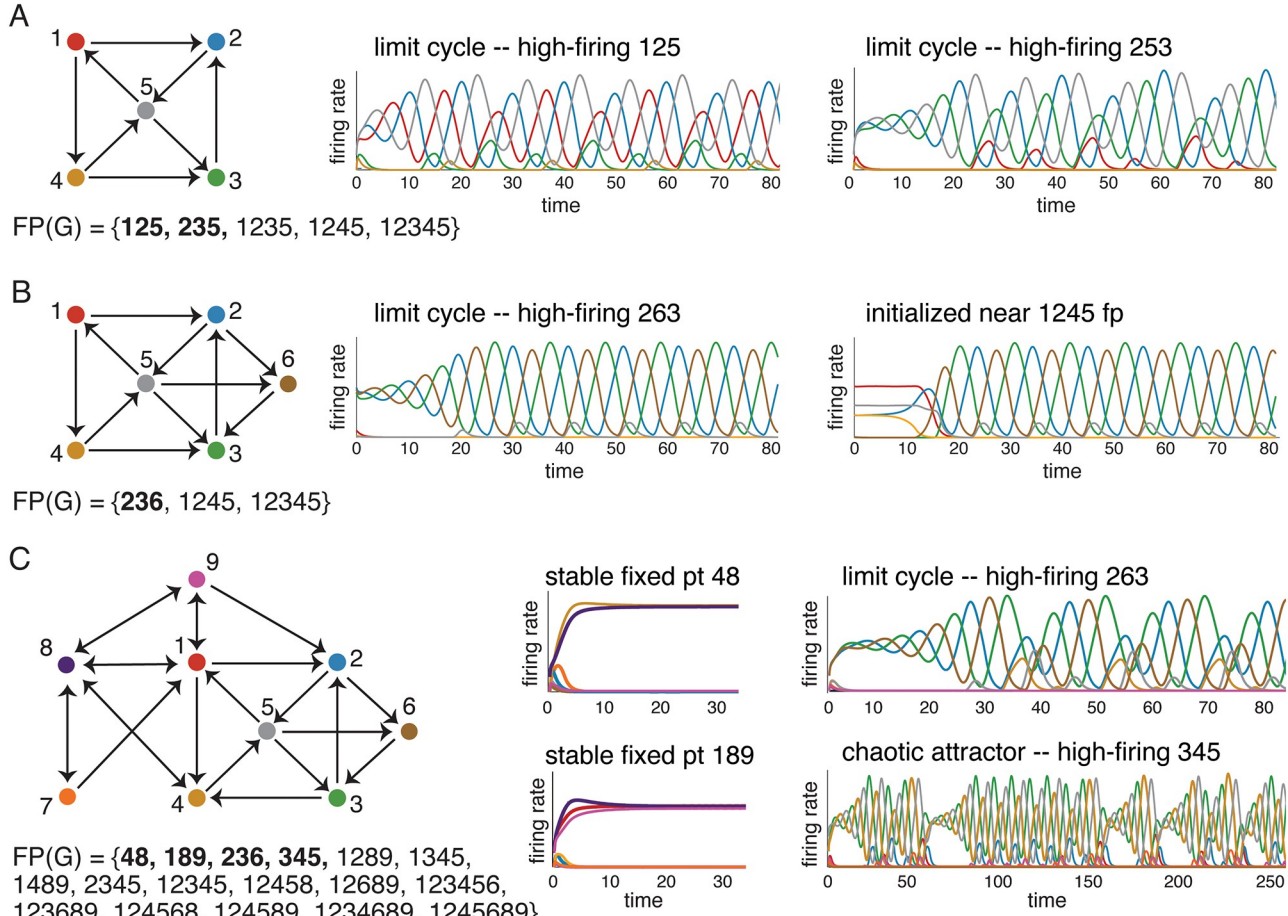

**Fig 4. Correspondence between core fixed points and attractors.** For each of the three graphs, FP(G) was computed using graph rules. Minimal fixed points that are also core fixed points are shown in bold. (A) A network on five nodes with two core fixed points supported on 125 and 235. Each of the two attractors of the network can be obtained via an initial condition that is a perturbation of one of these fixed points. The first attractor follows the cycle 125 in the graph, while the second one follows the cycle 253. (B) A network with the same graph as in A, except for the addition of node 6. Although there are two minimal fixed points, supported on 236 and 1245, only the fixed point for 236 is core and yields an attractor. Initial conditions near the 1245 fixed point (denoted 1245 fp) produce solutions that stay near the (unstable) fixed point for some time, but eventually converge to the same 236 attractor. (C) A larger network built by adding nodes 7, 8, and 9 to the graph in B, and flipping the 4 → 3 edge. This CTLN has four core fixed points, and no other minimal fixed points. Each core fixed point has a corresponding attractor: stable fixed points supported on 48 and 189, a limit cycle supported on 236, and a chaotic attractor for 345.

and also that it is minimal in $G$ if it survives to $\sigma \in FP(G)$. The converse, however, is not true. One can have $\sigma$ minimal in $G$ but not minimal in $G|_\sigma$, simply because the fixed points below it in the subnetwork did not survive to the larger one. This is what happened with the $\sigma = 1245$ fixed point in Fig 4B. These are precisely the minimal fixed points we are ruling out with the above definition.

Since core fixed points must satisfy $FP(G|_\sigma) = \{\sigma\}$, there are only certain subgraphs that can support them. It is useful to give these graphs their own name:

**Definition 2** (core motifs). Let $G|_\sigma$ be a subgraph of $G$ that satisfies $FP(G|_\sigma) = \{\sigma\}$. Then we say that $G|_\sigma$ is a *core motif* of $G$. When the graph in question is understood, we may also refer to the support itself, $\sigma$, as a core motif.

Using graph rules, it is easy to see that all cliques (all-to-all bidirectionally connected subgraphs) and cycles of any size are core motifs, and this holds for all choices of the CTLN

parameters $\varepsilon$, $\delta$, and $\theta$, provided they are within the legal range [13]. However, there is a richer variety of core motifs beyond cliques and cycles. Fig 5A–5C depicts all core motifs up to size $n = 4$. Note that all the graphs in Fig 5A and 5B are uniform in-degree, but not all are cliques or cycles. The second graph in Fig 5B has uniform in-degree 2 but no symmetry. Interestingly, its corresponding attractor exhibits a (2,3) exchange symmetry, as neurons 2 and 3 fire synchronously. Fig 5C shows the two core motifs of size $n \leq 4$ that are *not* uniform in-degree. The first one has what we call a *fusion attractor*, as it appears to be a blend of the usual limit cycle supported on 123 together with a fixed point supported on 4. The second graph in Fig 5C is an

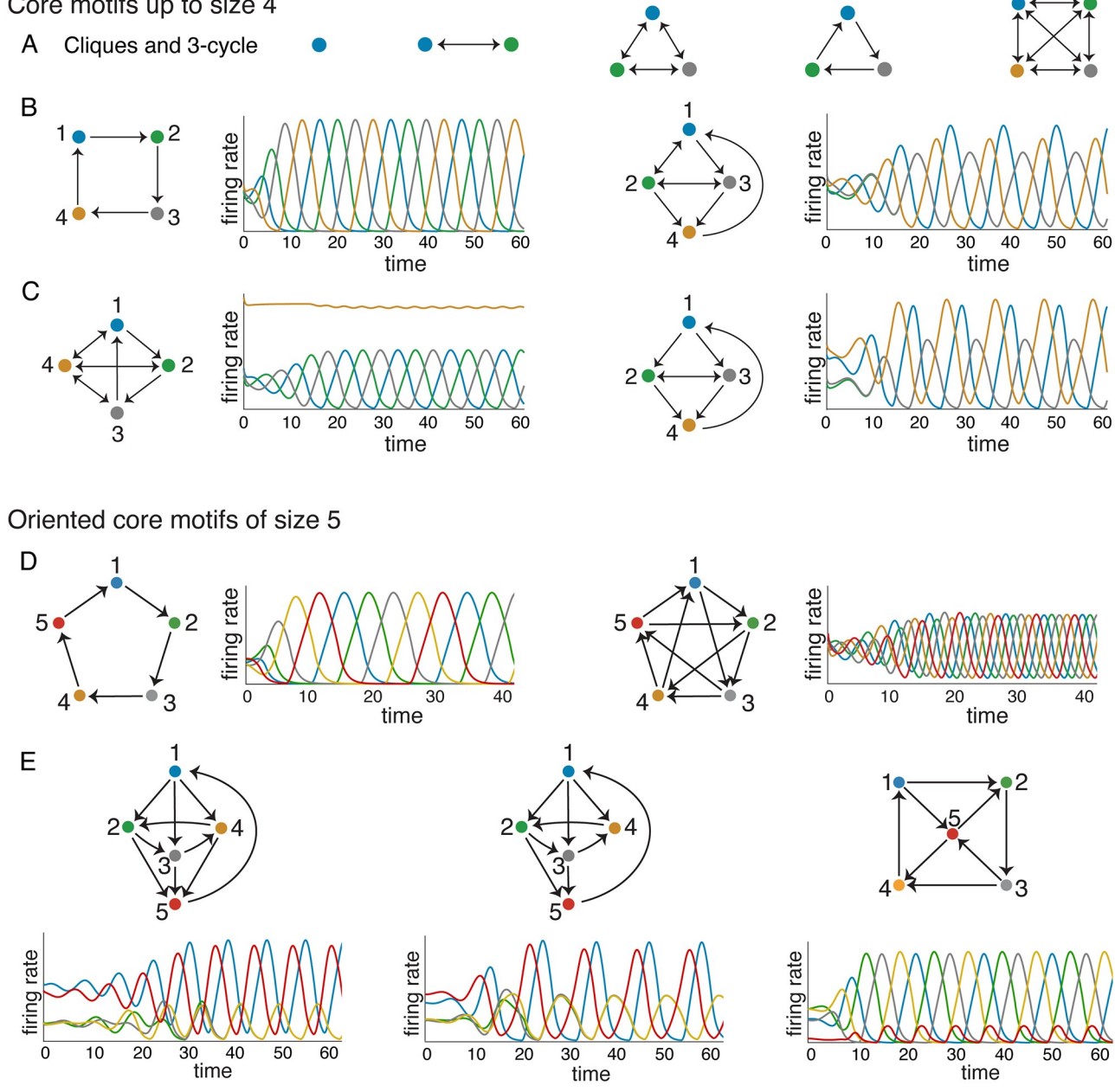

**Fig 5. Core motifs.** (A-C) All core motifs of size $n \leq 4$. Note that every clique is a core motif, as are all cycles. (B-C) Attractors are shown for each core motif of size 4 other than the 4-clique, whose attractor is a stable fixed point. (D-E) All $n = 5$ core motifs that are *oriented* graphs.

example of a *cyclic union* (see [11]), with a (2,3) exchange symmetry that is reflected in its attractor. The attractors for the cliques are all stable fixed points, and the 3-cycle attractor is the one we saw in Fig 1C.

There are many more core motifs of size $n$ = 5. Fig 5D–5E shows all the ones that are *oriented*, meaning the graphs have no bidirectional edges. Fig 5D shows the 5-cycle (left) and the 5-star (right), each having attractors that respect the cyclic symmetry. Fig 5E shows the remaining three oriented core motifs and their corresponding attractors. Note that the second attractor exhibits the same (2,3,4) symmetry as the first one, even though the dropped $4 \rightarrow 5$ edge breaks the corresponding symmetry in the graph.

Every core fixed point is supported on a core motif, but not all core motifs in a graph give rise to core fixed points. This is because the fixed point of a core motif may not survive the embedding in the larger network. For example, singletons are core motifs, but they only yield core fixed points if they are embedded as sinks in $G$. Similarly, all 3-cycles of a graph are core motifs, but they can fail to have a surviving fixed point as in the case of 145 in Fig 4A.

The graph in Fig 4C has plenty of cycles and cliques, and these are all core motifs. However, only the cliques supported on 48 and 189 and the 3-cycles supported on 236 and 345 have surviving fixed points in FP($G$). These are in fact all the core fixed points of $G$. Although FP($G$) has an additional 13 fixed points (shown in Fig 4C), none of them are minimal and so none can be core. By systematically trying a battery of different initial conditions in the state space, we were able to find only four attractors: two stable fixed points, a limit cycle, and a chaotic attractor. As can be seen in Fig 4C (right), these attractors correspond precisely to the four core fixed points we determined using graph rules. Moreover, each core fixed point was supported on a clique or a cycle core motif. In other words, by identifying the core motifs of the network and applying the uniform in-degree rule, we were able to find all core fixed points via a purely graphical analysis. These core fixed points were then predictive of the network's static and dynamic attractors.

We hypothesized that this pattern holds more generally: namely, that a network's core fixed points correspond to both its static and dynamic attractors. In the case of static attractors, our hypothesis implies that every stable fixed point is a core fixed point, and hence its support is minimal and corresponds to a core motif. In prior work, we explored an even stronger conjecture: that every stable fixed point of a CTLN corresponds to a target-free clique [12]. In this work, however, we test the complementary hypothesis that dynamic attractors correspond to core fixed points that are unstable.

## Attractor prediction

In order to focus on dynamic attractors, we decided to study graphs whose CTLNs contain no stable fixed points, and hence no static attractors. In [9] it was shown that oriented graphs with no sinks have no stable fixed points. We thus focused our attention on oriented graphs with no sinks on $n$ = 5 nodes. This family is large enough to encompass a rich variety of dynamic phenomena, and small enough to be studied comprehensively.

**Oriented graphs with no sinks.** A directed graph is *oriented* if it has no bidirectional edges $i \leftrightarrow j$. The graph in Fig 3A is oriented but has a sink, which corresponds to a static attractor. The graph in Fig 4C has no sinks, but it has bidirectional edges and is thus not oriented. It has both static and dynamic attractors. In contrast, the graphs in Fig 4A and 4B are both oriented with no sinks, as are the graphs in Fig 5D and 5E. These networks are guaranteed to have only dynamic attractors, and are precisely the kind of networks we have chosen to investigate. Note that their fixed points all have support of size at least three, a useful fact that holds more generally:

**Lemma 3**. *Let G be an oriented graph with no sinks. Then for each $\sigma \in \mathrm{FP}(G)$, $|\sigma| \geq 3$.*

*Proof.* By the sinks rule, there are no singletons in $\mathrm{FP}(G)$. Now consider $\sigma = \{i, j\}$, with $i \rightarrow j$. By the sources rule, $\sigma \notin \mathrm{FP}(G)$. On the other hand, if $\sigma = \{i, j\}$ but $G|_\sigma$ is an independent set (no edge), then by the uniform in-degree rule for $d = 0$ we see that $\sigma \notin \mathrm{FP}(G)$, since $i$ and $j$ are not sinks in $G$. As there are no subsets with bidirectional edges, it follows that $\sigma \notin \mathrm{FP}(G)$ for all $\sigma$ of size $|\sigma| \leq 2$.

For $n = 3$, there is only one oriented graph with no sinks: the 3-cycle. For $n = 4$, there are seven such graphs (up to isomorphism). Three of them are obtained by adding a proper source node to the 3-cycle, with one, two, or three outgoing edges. In addition to this, there is the 4-cycle and three more graphs obtained by adding a node to the 3 cycle that is not a source. We call these the D, E, and F graphs (see Fig 6A). In total, there are eight (non-isomorphic) oriented graphs with no sinks on $n \leq 4$ nodes. As before, $\mathrm{FP}(G)$ for each of these graphs could be derived and core fixed points identified via graphical analysis. We then verified the correspondence between core fixed points and attractors computationally, for CTLNs with the standard parameters. Fig 6 shows the attractors corresponding to the D, E, and F graphs, as well as that of the 3-cycle and 4-cycle (S graph). The three graphs obtained by adding a source node to the 3-cycle (not shown) have the same $\mathrm{FP}(G)$ and the same attractor as the 3-cycle. Only the F graph has more than one core fixed point, but each one yields its own attractor, as predicted. With so few graphs, however, this was not a strong test of our hypothesis.

Fig 6 contains an additional T graph (a.k.a. the "tadpole") that is oriented, but has a sink. We include it because it is useful in our method for classifying $n = 5$ oriented graphs with no sinks, described below. Only the dynamic attractor is shown, but there is also a stable fixed point supported on node 4. Indeed, the correspondence between core fixed points and attractors for this graph was already shown in Fig 3.

Note that each attractor in Fig 6A has a corresponding *sequence*, giving the order in which the neurons reach peak activity within a single period of the limit cycle. Underlined numbers correspond to low-firing neurons; for example, the sequence 123$\underline{4}$ for the D attractor indicates that node 4 has a low peak as compared to the other three. Since the trajectories for limit cycles are periodic, the sequence is also understood to be periodic. Our convention is to select the lowest-numbered high-firing neuron as the starting point.

**Taxonomy of n = 5 oriented graphs with no sinks.** For $n = 5$, there are many more oriented graphs with no sinks. We developed a complete taxonomy of these graphs after splitting them into two groups: graphs with at least one source, and graphs with no sources. There are 76 graphs in each group (up to isomorphism), for a total of 152 oriented graphs with no sinks.

The graphs with sources must have *proper* sources, since an isolated node is also a sink. Fig 6B shows the family of graphs obtained by adding node 4 and then node 5 as sources to the 3-cycle. The dashed lines indicate optional edges. Keeping in mind that each added source must have out-degree at least one, we count 30 graphs in this family, up to isomorphism. The accompanying attractor is identical for every graph, and matches that of the isolated 3-cycle in Fig 6A. The second family of graphs in Fig 6B comes from adding node 5 as a proper source to the D graph. Since the D graph has no symmetry, we get $2^4 - 1 = 15$ distinct graphs this way. Here, too, each attractor is identical and matches that of the isolated D graph above. Fig 6B shows the counts for all the other graph families obtained by adding a source. In total, there are 76 of them.

The remaining oriented graphs with no sinks have no sources. It turns out that, other than the 5-cycle, these can all be constructed from one of the D, E, F, T, or S base graphs in Fig 6A by adding node 5 with at least one incoming and at least one outgoing edge. We developed a simple notation for these constructed graphs, which is illustrated in Fig 6C. The notation uses the letter of the base graph followed by the nodes sending incoming edge(s) to node 5, and

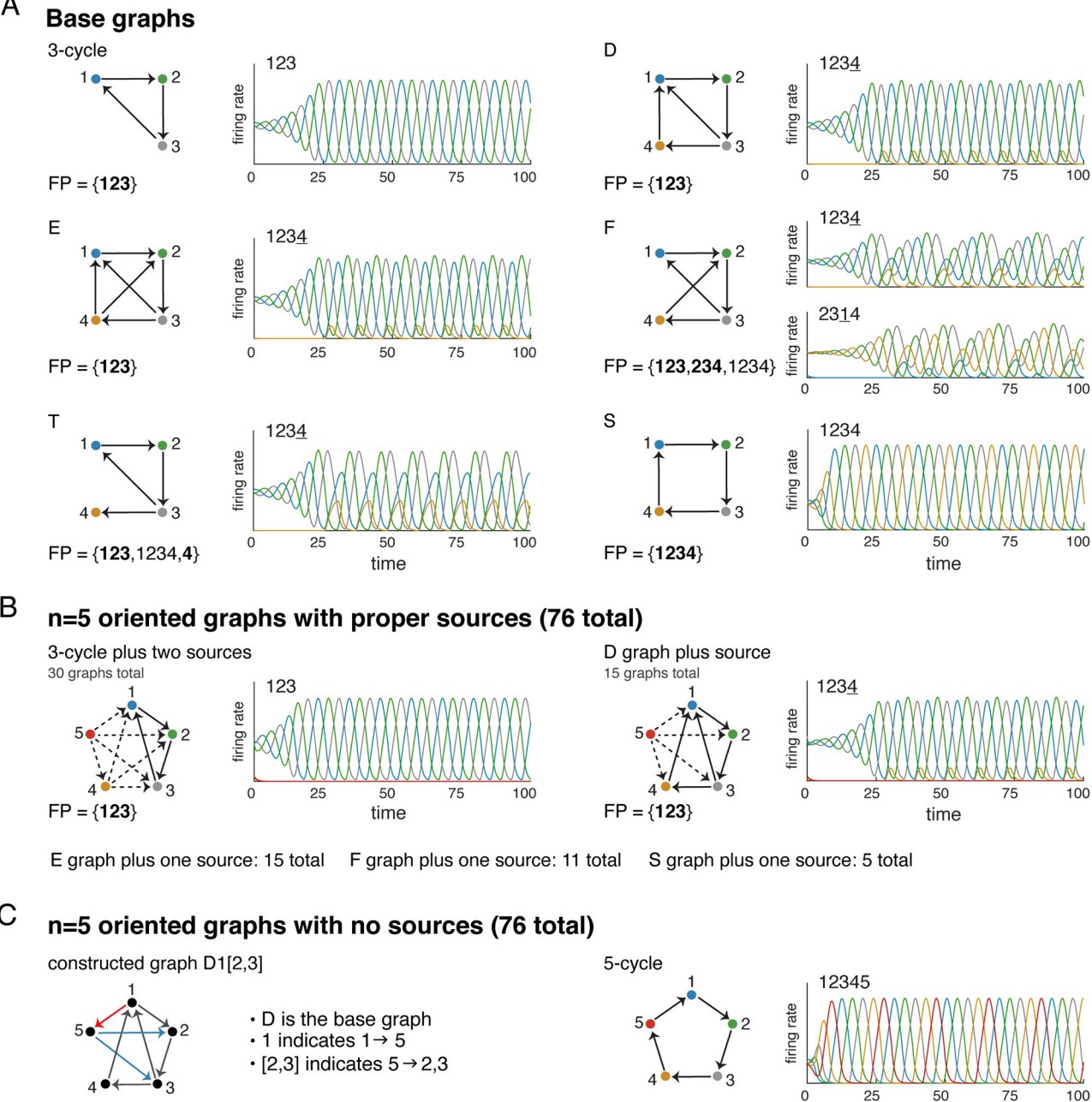

**Fig 6. Taxonomy of n = 5 oriented graphs with no sinks.** (A) Base graphs used to construct *n* = 5 graphs, and their corresponding attractors. Each attractor has a sequence, indicating the (periodic) order in which the neurons achieve their peak firing rates. (B) The oriented graphs with sources can be constructed by adding proper sources to each of the base graphs. This yields 30 graphs from the 3-cycle base (left), 15 graphs from the D graph base (right), and an additional 15, 11, and 5 graphs from the E, F and S graph bases. (C) All oriented graphs with no sources or sinks can be constructed from one of the D, E, F, T, and S base graphs. The graph label completely specifies the graph by naming the base and indicating the incoming and outgoing edges to the added node 5. (Left) For example, D1[2, 3] is the graph constructed from the D graph with added edges 1 → 5 and 5 → 2, 3. (Right) The only oriented *n* = 5 graph with no sources or sinks that cannot be constructed in this way is the 5-cycle.

finally the nodes receiving the outgoing edges from 5, in brackets. For example, the graph obtained from the D graph by adding the edges $1 \rightarrow 5$ and $5 \rightarrow 2, 3$ is denoted D1[2, 3]. (See Supporting information for more details.) All remaining graphs with no sources or sinks can be constructed in this manner, but many are constructible in more than one way. For example, D12, 3] is isomorphic to E2[3]. In total, there are 75 non-isomorphic graphs obtained via this construction. Together with the 5-cycle, these are precisely the 76 oriented graphs with no sources or sinks on $n = 5$ nodes.

**Parameter-independence of the attractor predictions.** From prior work [11], we know that for any oriented graph on $n \leq 5$ nodes, the set of fixed point supports FP($G$) is independent of the choice of CTLN parameters $\varepsilon$, $\delta$, and $\theta$, provided they are in the legal range. We were thus able to completely work out FP($G$) for each of the 152 graphs using graph rules, Lemma 3, and a few additional facts from [11] (see Supporting information). From this graphical analysis, we also identified the core fixed points for each graph. Altogether, there were 191 core fixed points across the 152 graphs, with at least one core fixed point per graph.

We hypothesized that each of these (unstable) core fixed points corresponds to a dynamic attractor, meaning that: (i) initial conditions near the fixed point yield solutions that converge to the attractor, and (ii) the support of the core fixed point predicts the high-firing neurons in the attractor. In other words, the core fixed points give us a concrete prediction for the attractors of the network, including how to find them. Note that while the exact firing rates at the core fixed points depend on parameters, their supports in FP($G$) are parameter-independent. This means the prediction for the number of attractors and where they are localized within the graph of the network is the same for all parameters $\varepsilon$, $\delta$, $\theta$ in the legal range. Table 2 provides a tally of the number of graphs and core fixed points for each subgroup of $n = 5$ graphs we studied.

**Testing the attractor predictions.** Next, we performed extensive searches for the attractors of the CTLNs associated to each graph, with the standard parameters. This involved simulations of network activity using a battery of initial conditions, including dozens of perturbations of each fixed point (not only core fixed points) and the 32 corners of the unit cube $[0, 1]^5$. Remarkably, we found that every single observed attractor corresponded to a core fixed point of the CTLN, and was thus accurately predicted by graphical analysis of the network. In particular, there were no *spurious* attractors that were not predicted by core fixed points. On the other hand, there were six core fixed points, across six different graphs, that did not have a corresponding attractor. We refer to these predicted attractors that failed to be realized as *ghost* or *missing* attractors. The results are summarized in Table 2.

The attractors were predicted irrespective of the CTLN parameters, but we initially tested these predictions only in the standard parameters. Since there were ghost attractors that failed to be realized, we wondered if these attractors might emerge in a different parameter regime. For each of the six graphs with ghost attractors, we investigated CTLNs with higher inhibition levels (i.e., higher $\delta$). We found that by keeping $\theta = 1$ and $\varepsilon = 0.25$, but increasing the inhibition to $\delta = 1.25$, all six ghost attractors were observed as real attractors in the network. Moreover, all the other core fixed points continued to have corresponding attractors, and there were no new spurious attractors (see Supporting information). In this parameter regime, the prediction of attractors from core fixed points was perfect.

**Modularity of attractors.** Our taxonomy of oriented graphs with no sinks allowed us to go further in our analysis of attractors. Other than the 5-cycle, each $n = 5$ graph was constructed from a base graph of three or four nodes, where the base graph contained a core motif embedded in a canonical way. So rather than starting from a set of 152 graphs with arbitrary orderings on the vertex labels, graphs with similarly embedded core motifs had their vertices aligned. This ensured that similarities across attractors corresponding to isomorphic core

**Table 2. Core fixed points and attractors for *n* = 5 oriented graphs with no sinks.** The attractors were found in CTLNs with the standard parameters.

| graphs | # graphs | # core fps | # attractors | # ghost atts | # spurious atts |
|---|---|---|---|---|---|
| with a source | 76 | 87 | 87 | 0 | 0 |
| with no source | 76 | 104 | 98 | 6 | 0 |
| total | 152 | 191 | 185 | 6 | 0 |

motifs were readily apparent, without having to find the optimal permutations on $x_1, \ldots, x_5$ to make the trajectories align. In particular, we expected that activation *sequences* associated to attractors for similarly embedded core motifs would be the same.

Beyond such combinatorial features, however, we expected considerable variation in the precise trajectories of the attractors. After all, no two graphs are isomorphic, so the core motifs across different graphs are never embedded in exactly the same way. In particular, for a given choice of CTLN parameters, each graph yields a distinct dynamical system having a distinct $W$ matrix, with no pair of matrices being permutation-equivalent.

To our surprise, we discovered that attractors from different networks corresponding to similarly embedded core motifs were often identical, or nearly identical. Moreover, the graphs fit into simple graph families which could be described compactly via a set of common edges across all graphs in the family, together with a set of optional edges that accounted for differences between graphs. We depict these families via *master graphs*, with optional edges shown as dashed lines. Although the optional edges could alter FP($G$), and even the number of attractors of a network, they left the aligned attractor for the graph family unchanged. Altogether we found that the 185 attractors observed in the standard parameters fell into only 25 distinct attractor classes, which we labeled att 1–25. The first attractor class, att 1, corresponds to the "pure 3-cycle" attractor shown in Fig 6B (left), with 30 graphs. While several attractor classes have only one graph, they vary considerably in size and the largest has 44 graphs. A full classification of attractor classes, together with master graphs, is provided in Section 4 of the Supporting Information.

Fig 7 displays eight of the attractor classes comprising 98 attractors and 2 ghost attractors. The first class shown, att 2, consists of all graphs obtained by adding a proper source to the 4-cycle. Note that since our graphs have no sinks, node 5 must have at least one outgoing edge. Although there appear to be $2^4 - 1 = 15$ possibilities, many of these are permutation-equivalent and the total count is only 5 graphs. Each of these graphs has a single attractor corresponding to the core fixed point supported on $\sigma = 1234$, as predicted. But the striking thing about these attractors is that they all appear to be identical: not only do they have the same sequence of activation, 1234, but the rate curves look exactly the same, matching the example shown in Fig 7 (att 2, top left). The second class, att 4, also has optional edges coming out of node 5. In this case, however, there is no symmetry, so all $2^3 - 1 = 7$ options produce non-isomorphic graphs. The reason there are 8 attractors of this type is that one of the graphs, corresponding to choosing only the $5 \rightarrow 3$ optional edge, has a symmetry that exchanges the 123 and 345 cycles. This results in a second attractor, with sequence 3145, that is isomorphic to the first one.

Attractor classes att 5 and att 6 have graphs that break into two families: one with 5 as a source node, and one where 5 receives a single edge from the base graph. Between them, they account for 44 + 29 = 73 of the 191 observed attractors. Graphs in att 5 are constructed from D and E base graphs, while the att 6 graphs all have an F graph as their base. The attractors for each family of graphs are the same, irrespective of whether the graph contains a source. For att 5, they match the isolated D and E attractors in Fig 6A, and for att 6 they match the isolated F attractor.

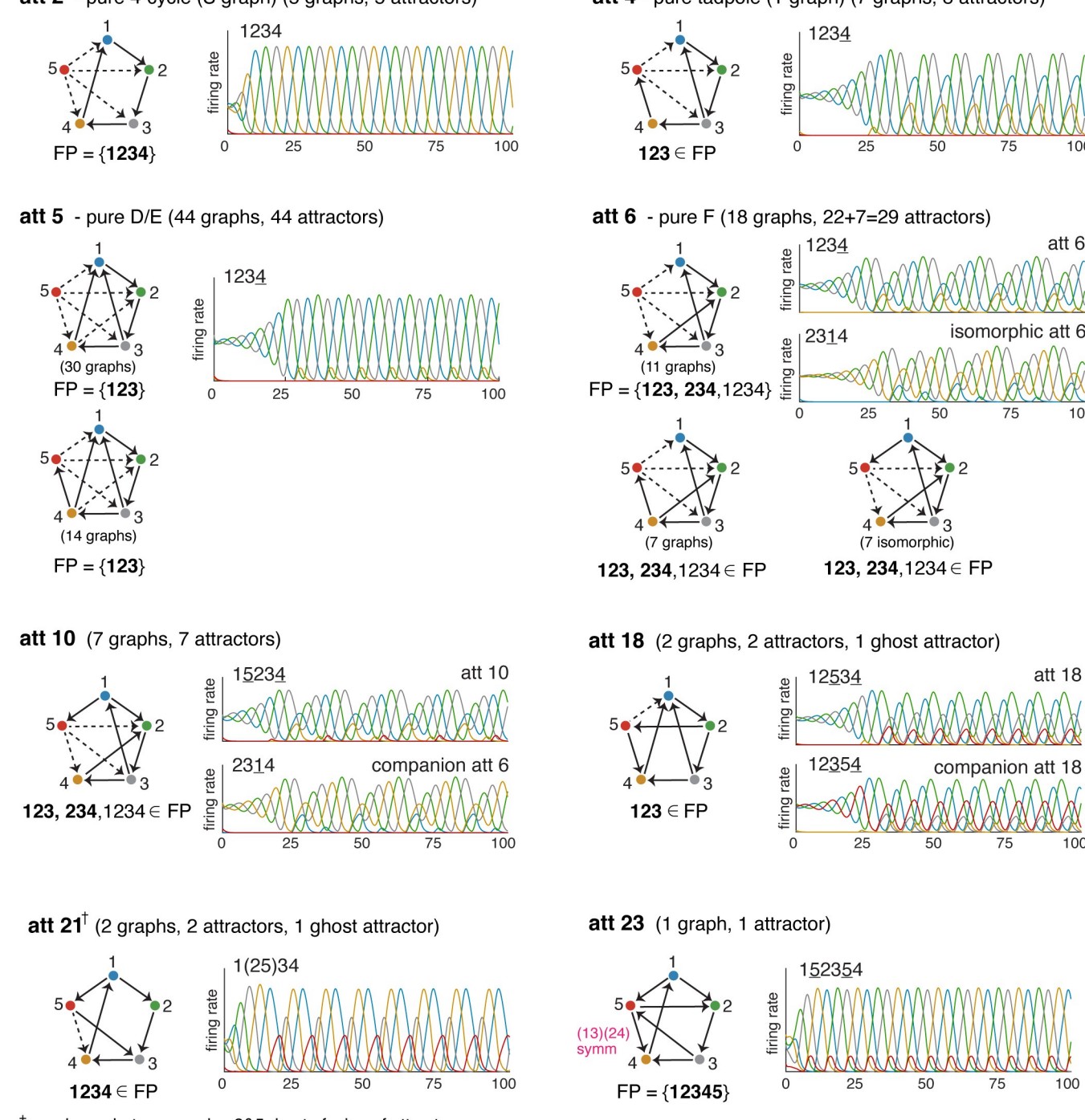

**Fig 7. Attractor classes and master graphs.** A sampling of attractor classes from the full classification for $n = 5$ oriented graphs with no sinks. Each attractor emerges from multiple graphs which, once properly aligned, fit neatly into families that can be summarized by "master graphs" with optional edges depicted via dashed lines. For families where FP($G$) is invariant across all graphs, the full form is shown. Otherwise, only the common fixed point supports are given. Some families always have two attractors: in these cases, the secondary attractor is shown as a "companion attractor" next to the relevant master graph. Note that the graph for att 23 has an automorphism, shown in pink. The full classification and further details of our notational conventions are provided in the Supporting Information.

Note that in att 5, the bottom graph family where 5 is not a source includes graphs where 234, 345, or 2345 is a core motif (a cycle). None of these cycles can have a core fixed point, however, because node 1 receives two edges from it. In fact, FP($G$) = {123} for all of these graphs and they each have a single attractor. In contrast, each graph under att 6 has a secondary attractor corresponding to 234. For the 11 graphs in the family where node 5 is a source, this is manifested as an isomorphic att 6. The remaining 7 graphs can be represented in two ways, depending on whether we have the $4 \rightarrow 5$ edge or the $1 \rightarrow 5$ edge (by symmetry, these are equivalent). In these cases, the secondary attractor is att 10, also shown in Fig 7. Note that although att 6 and att 10 are both supported on an embedded 3-cycle and have a similar appearance, the attractors are fundamentally different: att 6 does not involve node 5, while att 10 does include 5 as a low-firing node.

The last three classes shown in Fig 7 are att 18, att 21, and att 23. All three of these classes contain graphs with symmetry, and this affects the attractors in different ways. In att 18, when the $5 \rightarrow 1$ edge is present the graph has a (3, 5) exchange symmetry that exchanges the 123 and 125 core fixed points. Consequently, there are two isomorphic versions of att 18 in this graph. When $5 \nrightarrow 1$, this symmetry is broken, and the attractor for 123 disappears (another attractor for the 4-cycle 1245 emerges). We thus have a ghost attractor in the standard parameter regime. At the higher $\delta$ parameters, this attractor is realized and matches the pair for the other graph (see Supporting information).

Instead of exchanging multiple isomorphic attractors, symmetry can also fix an attractor. This can manifest itself in two different ways: the nodes exchanged by the symmetry may have synchronous activity in the attractor, or the attractor may display a time-translation symmetry, where permuting the nodes simply shifts a trajectory in time. The latter is the kind of symmetry we see in the isolated 3-cycle and 4-cycle attractors. In att 21, the graph without the $5 \rightarrow 4$ edge has a (2, 5) exchange symmetry, and this leads to nodes 2 and 5 firing synchronously in the attractor (see Fig 7, bottom left). The synchronous nodes are indicated by parentheses, so that the sequence is 1(25)34. At higher values of $\delta$, however, this synchrony is broken and the attractor actually splits into two isomorphic limit cycles that are exchanged by the (2, 5) symmetry (see Supporting information). On the other hand, att 23 has a symmetry that manifests itself as a time shift of the trajectory, fixing the attractor without any synchrony, similar to the 3-cycle case.

Table 3 gives a summary of the graph families and number of attractors for each of the 25 attractor classes. It also shows the sequence associated to each attractor. Note that different attractor classes may have the same sequence. For example, att 4, att 5, and att 6 each have the sequence 123$\underline{4}$. However, as can be seen in Fig 7, the limit cycles are visually quite different. In the Supporting Information, we provide a complete dictionary of all the oriented graphs with no sources and sinks, together with their corresponding attractors and sequences. We also exhibit the full set of attractor classes together with diagrams of their graph families, as in Fig 7.

Altogether, we have observed that in addition to core fixed points accurately predicting all the observed attractors, the embedded core motifs clustered into families of graphs that can be compactly described via a set of common and optional edges. All graphs in the same attractor class displayed identical or nearly identical attractors, even when the overall graph differed in important ways (including in the other attractors). This striking modularity of the attractors means that the same attractor, up to fine details of the dynamics, can be embedded in different networks whose dynamics may vary considerably otherwise.

**Failures of attractor prediction.**    For oriented graphs with no sinks up to $n$ = 5 nodes, we saw that all observed attractors were predicted by core fixed points. However, there were instances of "ghost" attractors where a core fixed point had no corresponding attractor in the

**Table 3. Graph families for attractor classes of $n = 5$ oriented graphs with no sinks.** The "$\sim$" notation indicates a forbidden edge, while *s indicate optional edges. For example, D2[$\sim$3,4,*] represents the pair of graphs D2[1, 4] and D2[4], which have no edge to node 3. See Supporting information for more details.

| attractor | sequence | graph families | # graphs | # attractors | # ghosts |
|---|---|---|---|---|---|
| att 1 | 123 | 3-cycle + sources | 30 | 30 | 0 |
| att 2 | 1234 | 4-cycle + source (aka S0[*]) | 5 | 5 | 0 |
| att 3 | 12345 | 5-cycle | 1 | 1 | 0 |
| att 4 | 123<u>4</u> | T4[*] | 7 | 8 | 0 |
| att 5 | 123<u>4</u> | D/E0[*] & D/E4[*] | 30+14 = 44 | 30+14 = 44 | 0 |
| att 6 | 123<u>4</u> | F0[*] & F4[*] | 11+7 = 18 | 22+7 = 29 | 0 |
| att 7 | 15<u>2</u>34 | D1[2,*], E1[*], & D/E[1, 4][2,*] | 4+7+4 = 15 | 4+7+4 = 15 | 0 |
| att 8 | 123(45) | D/E3[$\sim$4,*] | 5 | 5 | 0 |
| att 9 | 123<u>54</u> | D/E3[4,*] | 8 | 8 | 0 |
| att 10 | 15<u>2</u>34 | F1[*] | 7 | 7 | 0 |
| att 11 | 12<u>5</u>34 | F2[3,*] & F[2, 4][3] | 3+1 = 4 | 4+1 = 5 | 1 |
| att 12 | 123<u>54</u> | F3[1,4,*] | 2 | 4 | 0 |
| att 13 | 123<u>54</u> | F3[$\sim$1,4,*] | 2 | 0 | 2 |
| att 14 | 123(45) | F3[2] | 1 | 3 | 0 |
| att 15 | 12<u>5</u>3<u>4</u> | F2[1, 4] | 1 | 4 | 0 |
| att 16 | 12<u>5</u>3<u>4</u> | E2[1, 4] | 1 | 2 | 0 |
| att 17 | 12<u>5</u>34 | E2[4] | 1 | 1 | 0 |
| att 18 | 12<u>5</u>34 | D2[$\sim$3,4,*] | 2 | 2 | 1 |
| att 19 | 15<u>23</u>4 | D1[4] | 1 | 1 | 0 |
| att 20 | 15234 | S1[4] | 1 | 1 | 0 |
| att 21 | 1(25)34 | S1[$\sim$2,3,*] | 2 | 2 | 1 |
| att 22 | 15<u>2</u>34 | S1[2,*] | 4 | 4 | 0 |
| att 23 | 15<u>2</u>354 | S[1, 3][2, 4] | 1 | 1 | 0 |
| att 24 | 23(154) | E[1, 3][4,*] | 2 | 2 | 0 |
| att 25 | 12534 | E[1, 2][3, 4] | 1 | 1 | 0 |

standard parameters, though these attractors were all observed in a higher $\delta$ regime. Fig 8A illustrates what happens when initial conditions are chosen near a core fixed point with a missing attractor. The graph D2[4] has two core fixed points, supported on 123 and 1245. When initial conditions are chosen near the 1245 fixed point, the solution quickly falls into the limit cycle with sequence 12(35)4, which is isomorphic to att 21 (top right). However, when initial conditions near the 123 fixed point are chosen, the activity initially spirals out with increasing amplitude in a 123 sequence, but does not settle on the corresponding attractor. Instead, the activity converges to the same attractor we saw before (bottom right). At higher $\delta$, however, the analogous initial condition does produce a different attractor (see Supporting information). Fig 8B shows a graph where the ghost attractor is almost viable. Here it is the core fixed point for 135 that has a missing attractor. Interestingly, initial conditions near this fixed point appear to converge to an attractor supported on 135 (bottom right). However, the solution is not stable and eventually the activity falls out of this pattern and converges to the attractor for 123 (top right).

Although we did have some prediction failures in the form of ghost attractors, every attractor we observed for the $n = 5$ oriented graphs with no sinks was predicted by a core fixed point. There were no "spurious" attractors (see Table 2). It turns out, however, that spurious attractors can also occur as a failure of prediction. Fig 8C shows three graphs that have no core

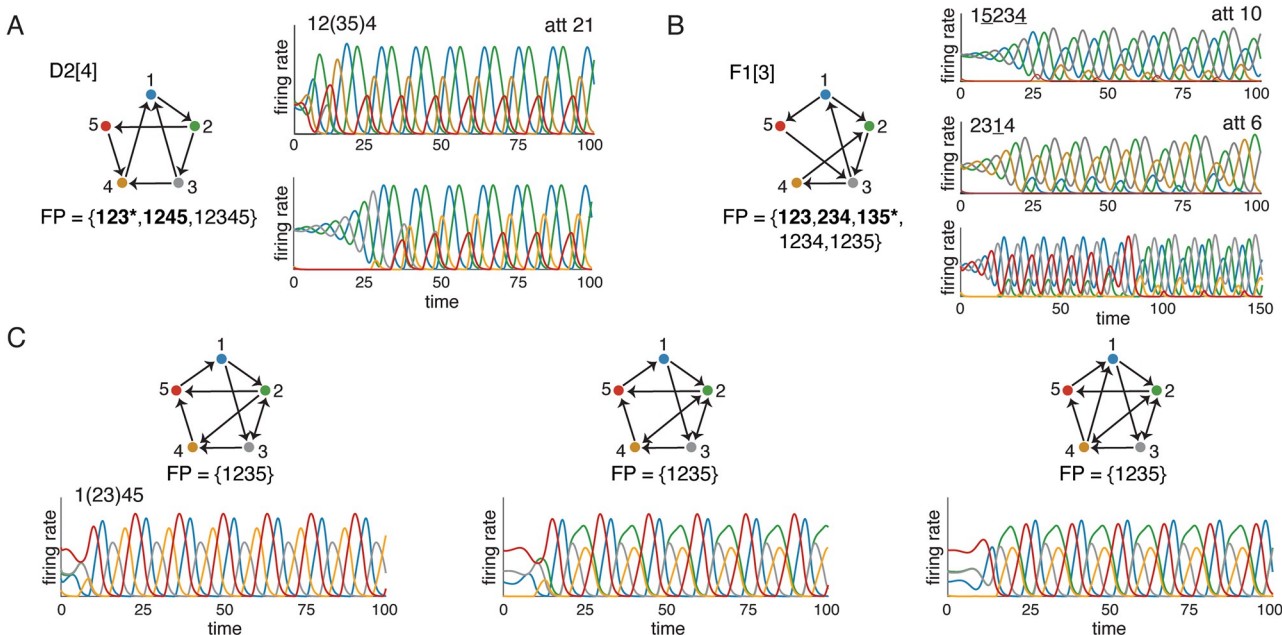

**Fig 8. Failures of attractor prediction from core fixed points.** (A) The graph D2[4] has two core fixed points, but only one attractor (att 21, top right). Initializing near the core fixed point with support 123 leads to activity that eventually falls into the 1245 attractor (bottom right). (B) The graph F1[3] has three core fixed points, but only the first two have corresponding attractors. Initializing near the fixed point for 135 initially appears to fall into an attractor supported on 135 (bottom right). However, after time these solutions converge to the attractor supported on 123. The missing attractors in A-B are called "ghost attractors." In a higher $\delta$ parameter regime, however, the core fixed points do yield their own attractors (see Supporting information). (C) Three graphs that are *not* oriented: each one has the bidirectional edge 2 ↔ 3. These graphs each have a unique fixed point, supported on 1235, but it is not a core fixed point. Nevertheless, the corresponding networks all have dynamic attractors.

fixed points, but still exhibit a limit cycle attractor. These are all, by definition, spurious attractors. Note that each graph contains at least one bidirectional edge, 2 ↔ 3, and so these graphs are all outside our oriented graphs family. In each case, FP(G) = {1235}, and this corresponds to an F graph. It is not a core motif because the F graph has three fixed points (the two minimal ones do not survive the embedding).

Finally, Fig 9 shows that symmetry can also lead to spurious attractors. This time the problem is not that there is no core fixed point, but that the same core fixed point corresponds to more than one attractor. Fig 9A shows that this can happen even with one of our oriented $n = 5$ graphs, the 5-star. Specifically, if we go to a different parameter regime (in this case $\varepsilon = 0.1$, $\delta = 0.12$), we find that some perturbations of the core fixed point lead to the expected attractor with sequence 12345, while other initial conditions obtained by perturbing from the same fixed point lead to a very different and unusual attractor (bottom). Fig 9B shows that a similar phenomenon occurs on the cyclically symmetric tournament on $n = 7$ nodes. Here, we see two very distinct solutions corresponding to the same (and only) core fixed point. The top solution is a limit cycle, and the bottom one is quasiperiodic. The projection of the trajectories (bottom left) shows the fixed point and limit cycle in red, and the quasiperiodic trajectory with toroidal structure in black. In both networks, the graph is highly symmetric and this symmetry seems to give rise to the additional "spurious" attractors.

## Discussion

Predicting dynamic attractors from network structure is notoriously difficult. In this work, we have shown that in the case of CTLNs, the problem is surprisingly tractable. Specifically, we

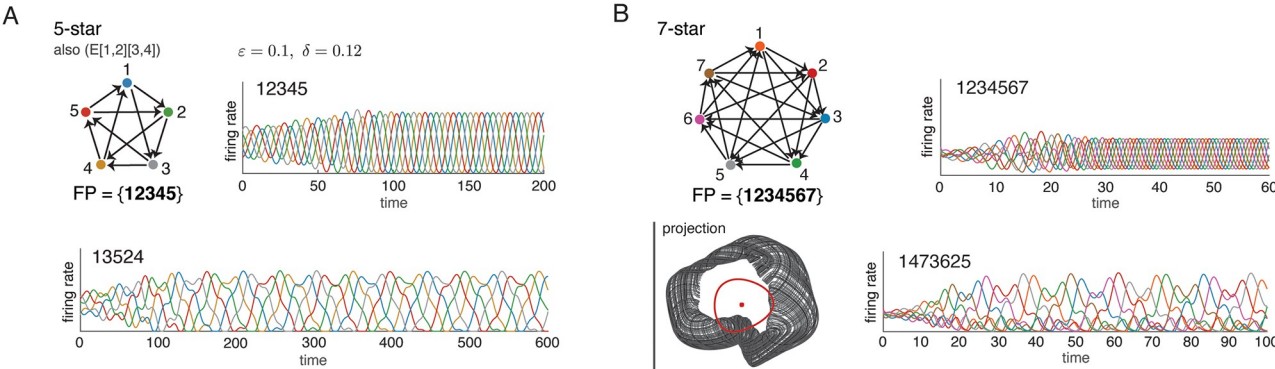

**Fig 9. Symmetry can lead to spurious attractors.** (A) Although the 5-star graph has only a single attractor in the standard CTLN parameters, for $\varepsilon = 0.1$, $\delta = 0.12$ a second attractor emerges (bottom). Both can be accessed via small perturbations of the unique fixed point. (B) The 7-star graph also has two attractors that can be accessed from a single core fixed point, even in the standard parameters. The projection (bottom left) depicts a random projection of $\mathbb{R}^7$ onto the plane, with trajectories for the limit cycle (red circle) and an additional quasiperiodic attractor (black torus). The fixed point is also shown (red dot).

have observed a correspondence between the core fixed points of a network and its attractors. Moreover, these core fixed points have minimal supports in FP(G) and correspond to special graphs called core motifs. Using graph rules, it is straightforward to identify core fixed points in small CTLNs via structural properties of the network.

We hypothesized that core fixed points can be used to predict attractors in CTLNs. The prediction is that for each core fixed point, there is an attractor whose high-firing neurons correspond to the support of the fixed point, and the attractor can be accessed by initial conditions that are small perturbations of the fixed point. In the case of stable fixed points, which arise for core motifs that are cliques, this prediction trivially holds. So we set out to test the hypothesis for unstable core fixed points, which give rise to dynamic attractors. We focused on oriented graphs with no sinks, as these networks are guaranteed to have only unstable fixed points.

Out of 152 oriented graphs with no sinks on $n = 5$ nodes, we observed 185 attractors. All of them were predicted by core fixed points. Moreover, we found that the attractors clustered into only 25 attractor classes, with attractors in the same class being nearly identical. We were also able to organize the graphs having the same attractor into simple structural graph families. These graph families highlight the close connection between structural properties of embedded core motifs and the resulting dynamic attractors. In particular, the same attractor can be embedded in different networks whose dynamics are completely different outside the common attractor.

Our attractor prediction was not perfect, however. In the standard parameter regime, we also had 6 failures in the form of ghost attractors, which were predicted but not realized by the network. We also saw examples of networks with no core fixed points, that nevertheless had dynamic attractors. Finally, we observed that highly symmetric networks can have core fixed points that give rise to multiple attractors. Despite these caveats, we conclude that core motifs and core fixed points are important tools for connecting network structure to dynamics. And in small networks, the attractor predictions from core fixed points are surprisingly accurate.

## Supporting information

**S1 Fig. Base graphs and graph counts.** (A) Base graphs used to construct $n = 5$ graphs, and their corresponding attractors. Each attractor has a sequence, indicating the (periodic) order

in which the neurons achieve their peak firing rates. (B) The oriented graphs with sources can be constructed by adding proper sources to each of the base graphs. This yields 30 graphs from the 3-cycle base (left), 15 graphs from the D graph base (right), and an additional 15, 11, and 5 graphs from the E, F and S graph bases. (C) All oriented graphs with no sources or sinks can be constructed from one of the D, E, F, T, and S base graphs. (Left) For example, D1[2, 3] is the graph constructed from the D graph with added edges $1 \rightarrow 5$ and $5 \rightarrow 2, 3$. (Right) The only oriented $n = 5$ graph with no sources or sinks that cannot be constructed in this way is the 5-cycle. (Same as Fig 6 in the main text).
(TIF)

**S2 Fig. Construction of oriented graphs from base graphs.** (Top) Starting with a D base, the graph D1[2, 3] is constructed by adding a node 5 together with incoming edge $1 \rightarrow 5$ (red) and outgoing edges $5 \rightarrow 2$ and $5 \rightarrow 3$ (blue). An isomorphic graph, E2[3], can be constructed from an E base. (Bottom) The graph S[1,3][2,4] has two incoming edges to node 5, given in the first set of brackets. This graph cannot be constructed from any base with only one edge into node 5.
(TIF)

**S3 Fig. Finding the name for a given oriented graph with no sinks.** The nodes with lowest in-degree are $k$, $\ell$, and $m$. However, removing $k$ results in a graph with no cycles that cannot match one of our base graphs. Removing $\ell$ (top) uncovers a D graph base, while removing $m$ (bottom) results in an E base. The original graph can thus be labeled as D3[1, 2] or E3[1].
(TIF)

**S4 Fig. D graphs: D1[*] & D2[*].**
(TIF)

**S5 Fig. D graphs: D3[*] & D4[*].**
(TIF)

**S6 Fig. E graphs: E1[*] & E2[*].**
(TIF)

**S7 Fig. E graphs: E3[*] & E4[*].**
(TIF)

**S8 Fig. F graphs: F1[*] & F2[*].**
(TIF)

**S9 Fig. F graphs: F3[*].**
(TIF)

**S10 Fig. T and S graphs: T4[*] & S1[*].**
(TIF)

**S11 Fig. Constructed core motifs.**
(TIF)

**S12 Fig. Graphs with parameter-dependent attractors.**
(TIF)

**S13 Fig. Fig 4: Master graphs and their corresponding graph families.**
(TIF)

**S14 Fig. Attractor classes att 1–10.**
(TIF)

**S15 Fig. Attractor classes att 11–19.**
(TIF)

**S16 Fig. Attractor classes att 20–25.**
(TIF)

**S1 File. Taxonomy of attractors for oriented graphs on $n$ = 5 nodes.** This supplementary text describes a classification scheme for oriented graphs with no sinks on $n$ = 5 nodes. We also give a classification for all attractors of these graphs with standard parameters, and provide a dictionary of the attractor classes.
(PDF)

# Acknowledgments

We thank Carolyn Shaw for earlier contributions to the analysis of fixed points of CTLNs, which helped motivate the concept of core fixed points.

# Author Contributions

**Conceptualization:** Caitlyn Parmelee, Katherine Morrison, Carina Curto.

**Formal analysis:** Caitlyn Parmelee, Samantha Moore, Katherine Morrison, Carina Curto.

**Funding acquisition:** Katherine Morrison, Carina Curto.

**Investigation:** Caitlyn Parmelee, Samantha Moore, Katherine Morrison, Carina Curto.

**Methodology:** Caitlyn Parmelee, Katherine Morrison, Carina Curto.

**Software:** Katherine Morrison.

**Supervision:** Katherine Morrison, Carina Curto.

**Writing – original draft:** Carina Curto.

**Writing – review & editing:** Caitlyn Parmelee, Katherine Morrison.

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
