## [Decision Letter · Decision Letter 0]

7 Dec 2021

PONE-D-21-30804Core motifs predict dynamic attractors in combinatorial threshold-linear networksPLOS ONE

Dear Dr. Curto,

Thank you for submitting your manuscript to PLOS ONE.  We invite you to submit a revised version of the manuscript that addresses the points raised during the review process.

We look forward to receiving your revised manuscript.

Kind regards,

Ivan Kryven

Academic Editor

PLOS ONE

Journal Requirements:

“This work was supported by NIH R01 EB022862, NIH R01 NS120581, NSF DMS-1951165, and NSF DMS-1951599.”

“This work was supported by NIH R01 EB022862 (CC & KM), NIH R01 NS120581, NSF (CC), DMS-1951165 (CC), and NSF DMS-1951599 (KM). The funders had no role in study design, data collection and analysis, decision to publish, or preparation of the manuscript.”

Reviewers' comments:

Reviewer's Responses to Questions

**Comments to the Author**

1. Is the manuscript technically sound, and do the data support the conclusions?

Reviewer #1: Yes

Reviewer #2: Yes

2. Has the statistical analysis been performed appropriately and rigorously? 

Reviewer #1: N/A

Reviewer #2: N/A

3. Have the authors made all data underlying the findings in their manuscript fully available?

Reviewer #1: Yes

Reviewer #2: Yes

4. Is the manuscript presented in an intelligible fashion and written in standard English?

Reviewer #1: Yes

Reviewer #2: Yes

5. Review Comments to the Author

Reviewer #1: The authors of the paper study attractors in combinatorial Threshold-linear networks (CTLNs). Specifically, they investigate to what degree dynamic attractors in such systems correspond to so-called core fixed points.

CTLNs are a particular type of piecewise smooth neural network. It can be shown that (under some non-degeneracy conditions) fixed points can be labelled by subsets of the set of nodes. Core fixed points are then defined by certain minimality conditions on these subsets. A fixed point corresponds to an attractor if initial conditions close to the fixed point lead to the attractor in forward time, and if the dynamical behavior on the attractor involves strong activity exactly in the nodes of the subset corresponding to the fixed point. The hypothesis under investigation is that the core fixed points are precisely those that correspond to dynamic attractors.

This hypothesis is verified numerically on the set of all 5-node networks without sinks and without bi-directed links. This set is chosen because is it reasonably large (around 150 networks, up to isomorphic ones) and because they have only dynamic (as opposed to static) attractors. The results are in excellent agreement with the hypothesis, suggesting something intriguing is indeed going on. In addition, the attractors found can be classified in a relatively small number of cases, that furthermore agree to a very large extend with how the different networks can be built from certain set motifs.

It is my understanding that this paper represents a significant discovery concerning the extremely challenging search for the relation between interaction structure and dynamical behavior. It is furthermore written very well and can be read without much knowledge of the topic.

Some things to take into consideration are that I am not an expert in threshold linear networks, and am therefore not in a position to make a good judgement of the significance within this area. Moreover, (almost) all new results are numerical, and counter-examples of various sorts to the hypothesis are found. These are rare exceptions though, and furthermore analyzed in great detail. For instance, a different parameter regime is enough to locate missing attractors, and sometimes symmetry seems to create anomalous results (as is often the case in dynamical systems). It depends on journal policy if articles with solely numerical results are accepted. Rigorous results pertaining to the hypothesis might furthermore be very hard to prove (as is notorious for relations between network structure and dynamics, and considering the outliers found). Finally, there is always a risk that the observations found pertain only to networks with relatively small numbers of nodes. An example with a significantly larger number of nodes is presented though, where the hypothesis seems to hold.

In general, the results presented are very interesting. They are promising, not only in the field of TLNs, but also regarding the much wider field of network dynamical systems.

Some minor things:

Main document:

p. 2, l. 49: I realize terminology differs from field to field, but should "linear" not be "affine" here? This applies to more places.

p. 2, Figure 1: The sketch of [.]_+ in panel B is very minimalistic. Some information on the axis might make this a lot clearer.

p.3, l. 76: [n] is already used on line 71, and so should probably be defined already there instead.

p.3, l. 79: "is also fixed point"  "is also a fixed point"

p.3, l. 79: It becomes clear from context, but nevertheless it might be a good idea to shortly define subnetwork. I.e. they can have both incoming and outgoing arrows in the definition of the paper, it seems.

p.4, l. 97: "corresponding to single"  "corresponding to a single"

p.5, l. 151: How could a fixed point that does not survive an embedding have a corresponding attractor?

p.6, Figure 4: the last picture in panel B has a stray "fp" (whereas the others do not).

p.6, Figure 4: The reader learns only later why it says "263", instead of "236". A small mention that this is intensional and will be explained later might already clear it up.

p.11, l. 281: "oriented graph on n<=5, the set..."  "oriented graph on n<=5 nodes, the set..."

p.11, l. 293: "and where they are localized within the network is the same" what is meant here? certainly exact numerical details will differ when parameters are changed?

p.12, l. 358: "F attractor"  "F attractors" (?)

Supp. Mat.:

p. 1: "We have verified that, with the exception of the 5-cycle, all of these graphs can be constructed by adding a single vertex to one of the five base graphs D, E, F, T, S, or the 3-cycle, shown in Figure 1A." This sentence implies you can get a graph with 5 nodes by adding a single vertex to the 3-cycle.

p.17: "and the F graph as a (1,4) symmetry"  "and the F graph has a (1,4) symmetry"

Reviewer #2: Review uploaded as attachment due to use of mathematical typesetting.

6. PLOS authors have the option to publish the peer review history of their article (what does this mean?). If published, this will include your full peer review and any attached files.

Reviewer #1: No

Reviewer #2: No

---

## [Author Response · Author response to Decision Letter 0]

26 Jan 2022

We have included a PDF with a complete "response to reviewers" file in our resubmission.

---

## [Editor Report · Decision Letter 1]

11 Feb 2022

Core motifs predict dynamic attractors in combinatorial threshold-linear networks

PONE-D-21-30804R1

Dear Dr. Curto,

We’re pleased to inform you that your manuscript has been judged scientifically suitable for publication and will be formally accepted for publication once it meets all outstanding technical requirements.

Kind regards,

Ivan Kryven

Academic Editor

PLOS ONE
---

## [Editor Report · Acceptance letter]

24 Feb 2022

PONE-D-21-30804R1 

Core motifs predict dynamic attractors in combinatorial threshold-linear networks 

Dear Dr. Curto:

I'm pleased to inform you that your manuscript has been deemed suitable for publication in PLOS ONE. Congratulations! Your manuscript is now with our production department. 

Kind regards, 

on behalf of

Dr. Ivan Kryven 

Academic Editor

PLOS ONE